# The Osteoblast Transcriptome in Developing Zebrafish Reveals Key Roles for Extracellular Matrix Proteins Col10a1a and Fbln1 in Skeletal Development and Homeostasis

**DOI:** 10.3390/biom14020139

**Published:** 2024-01-23

**Authors:** Ratish Raman, Mishal Antony, Renaud Nivelle, Arnaud Lavergne, Jérémie Zappia, Gustavo Guerrero-Limón, Caroline Caetano da Silva, Priyanka Kumari, Jerry Maria Sojan, Christian Degueldre, Mohamed Ali Bahri, Agnes Ostertag, Corinne Collet, Martine Cohen-Solal, Alain Plenevaux, Yves Henrotin, Jörg Renn, Marc Muller

**Affiliations:** 1Laboratory for Organogenesis and Regeneration (LOR), GIGA Institute, University of Liège, 4000 Liège, Belgium; ratish.raman@uliege.be (R.R.); mishalantony88@gmail.com (M.A.); rnivelle4550@gmail.com (R.N.); g.guerrero.limon@pm.me (G.G.-L.); joergrenn@yahoo.de (J.R.); 2GIGA Genomics Platform, B34, GIGA Institute, University of Liège, 4000 Liège, Belgium; arnaud.lavergne@uliege.be; 3MusculoSKeletal Innovative Research Lab, Center for Interdisciplinary Research on Medicines, University of Liège, 4000 Liège, Belgiumyhenrotin@uliege.be (Y.H.); 4Hospital Lariboisière, Reference Centre for Rare Bone Diseases, INSERM U1132, Université de Paris-Cité, F-75010 Paris, France; cakaetano@hotmail.com (C.C.d.S.); agnes.ostertag@inserm.fr (A.O.); corinne.collet@aphp.fr (C.C.); martine.cohen-solal@inserm.fr (M.C.-S.); 5Laboratory of Pharmaceutical and Analytical Chemistry, Department of Pharmacy, CIRM, Sart Tilman, 4000 Liège, Belgium; priyanka.kumari@uliege.be; 6Department of Life and Environmental Sciences, Università Politecnica delle Marche, Via Brecce Bianche, 60131 Ancona, Italy; jerry.maria.sojan@gmail.com; 7GIGA CRC In Vivo Imaging, University of Liège, Sart Tilman, 4000 Liège, Belgium; christian.degueldre@uliege.be (C.D.); m.bahri@uliege.be (M.A.B.); annick.claes@uliege.be (A.P.); 8UF de Génétique Moléculaire, Hôpital Robert Debré, APHP, F-75019 Paris, France

**Keywords:** zebrafish, osteoblast, transcriptome, gene expression, *col10a1a*, *fbln1*, FGF8, ECM, skeletal development, vertebra

## Abstract

Zebrafish are now widely used to study skeletal development and bone-related diseases. To that end, understanding osteoblast differentiation and function, the expression of essential transcription factors, signaling molecules, and extracellular matrix proteins is crucial. We isolated Sp7-expressing osteoblasts from 4-day-old larvae using a fluorescent reporter. We identified two distinct subpopulations and characterized their specific transcriptome as well as their structural, regulatory, and signaling profile. Based on their differential expression in these subpopulations, we generated mutants for the extracellular matrix protein genes *col10a1a* and *fbln1* to study their functions. The *col10a1a^−/−^* mutant larvae display reduced chondrocranium size and decreased bone mineralization, while in adults a reduced vertebral thickness and tissue mineral density, and fusion of the caudal fin vertebrae were observed. In contrast, *fbln1^−/−^* mutants showed an increased mineralization of cranial elements and a reduced ceratohyal angle in larvae, while in adults a significantly increased vertebral centra thickness, length, volume, surface area, and tissue mineral density was observed. In addition, absence of the opercle specifically on the right side was observed. Transcriptomic analysis reveals up-regulation of genes involved in collagen biosynthesis and down-regulation of Fgf8 signaling in *fbln1^−/−^* mutants. Taken together, our results highlight the importance of bone extracellular matrix protein genes *col10a1a* and *fbln1* in skeletal development and homeostasis.

## 1. Introduction

Bone formation and homeostasis is a highly regulated process, whose better understanding will provide insights into diagnosis and therapeutic interventions to be developed for the welfare of an aging human population suffering from skeletal pathologies like osteoporosis, osteopetrosis, osteoarthritis, and age-/sports-related injuries. The two cell types critical for bone formation are the chondrocytes secreting the cartilage extracellular matrix (ECM) and the osteoblasts that are responsible for building new bone tissue [1]. Cartilage and bone cell fate is governed by the foremost regulator RUNX2 [2], whereas SP7 promotes osteoblast commitment in early osteoblast progenitors [3,4]. Together with these two cell types, osteocytes form the cellular component of the bone skeleton, ultimately responsible for deposition, mineralization, and maintenance of the bone ECM. The ECM forms the non-cellular component of the bone tissue, comprising the organic component made up of predominantly collagens and non-collagenous proteins, and the inorganic component consisting of calcium phosphate apatite and trace elements [5].

Zebrafish (*Danio rerio*) has recently become a powerful model for skeletal biology, due to its conserved genetic and functional characteristics compared to mammals and its numerous technical and experimental advantages [6,7,8]. The most studied skeletal structures during early development are the cranial bones, as they are the first to undergo ossification and are specially interesting for the study of craniofacial disorders [9]. Similar to terrestrial vertebrates, the skeleton of zebrafish contains bones of both dermal and chondral origins, which form from neural crest-derived cells relatively early in the course of development [10,11]. In later stages and in adults, the vertebral column and fin rays are extensively studied [12]. Individual structures of the skeleton are formed either by osteoblasts derived from mesenchymal cells, without a previous cartilage matrix (intramembranous), or through endo- or peri-chondral ossification building up onto a preformed cartilage extracellular matrix (ECM), previously secreted by chondrocytes [6,13]. Both the key regulators and signaling pathways controlling skeletal development are highly conserved between mammals and teleosts [14,15,16]. Indeed, zebrafish osteoblasts express specific genes [17], often orthologs of their mammalian counterparts, such as those for the transcription factors Runx2 (*runx2a* and *runx2b*) and Sp7 (*sp7*). The *sp7* gene is the earliest regulator and marker for osteoblasts in zebrafish [18].

The ECM plays a crucial role in the formation of the skeleton, not only as a protein backbone for the mineralization to form the bone tissues, but also through its regulatory interactions with the bone cells. The first zebrafish model for osteogenesis imperfecta (the *chihuahua* mutant) has a dominant mutation in the *col1a1a* gene, illustrating how modifying a major component of the bone ECM is able to affect skeletal development [19] and how such a model can help to better understand bone mineralization [20]. Other collagens have been shown to affect skeletal formation, such as Col2a1a, Col11a2 [21], or Col8a1a [22]. COL10A1 is interesting, as it is expressed in hypertrophic chondrocytes in fish and mammals, but only zebrafish (and other teleosts) express it in osteoblasts [23,24,25]. It is a nonfibrillar collagen forming a homotrimer of three identical chains. In mammals, COL10A1 plays a significant role in endochondral bone development due to its specific expression in hypertrophic chondrocytes, mainly in the calcifying zone of growth plate cartilage [26]. It is also expressed in the calcified zone of knee articular cartilage, where increased hypertrophy and COL10 expression are well documented in osteoarthritis [27]. In humans, missense, nonsense, and frame-shift mutations in the *COL10A1* gene cause Schmid type metaphyseal chondrodysplasia (SMCD) [28,29]. In mice, the presence of abnormal COL10A1 resulting from dominant acting mutations in the *Col10a1* gene affects trabecular bone and causes *cox vara*, reduced thickness of the growth plate resting zone and articular cartilage, altered bone content, and atypical distribution of matrix components within growth plate cartilage [30,31]. In zebrafish, however, the significance of this additional expression is unknown.

Genes coding for non-collagenous ECM proteins were also identified. The *bglapl* gene is often used as a marker for mature or late osteoblasts [32]. Other genes, such as *spp1* [33] or *gpc4* [34], were shown to be required for bone formation, in part by interacting with extracellular signaling proteins involved in WNT, BMP, or HH pathways that are crucial for skeletal formation. Another family of ECM proteins is the Fibulin family, glycoproteins that are found in basement membranes, fibers, and proteoglycan aggregates [35] in many locations where they play a role in organogenesis by stabilizing the ECM through their interactions with binding partners [36]. Its founding member, FBLN1, was shown to be required for bone mineralization in a mouse KO model [37]. Indeed, *Fbln1* KO results in defects in neural crest cell patterning, leading to anomalies of the aortic arch arteries, thymus, thyroid, cranial nerves, hemorrhages in the head and neck, and finally increased mortality [38,39]. Furthermore, *Fbln1* KO mice display a significant reduction in mineralization followed by reduced bone volume and size in the calvarial and frontal bones, at least in part by impeding *Sp7* induction by BMP2 [37]. In zebrafish, *fbln1* expression was observed in posterior presomitic mesoderm, the tail tip, and regions of somite formation. At later stages, it is expressed in fin mesenchymal cells [40] and in the myocardium [41]; however, its role in skeletal development of zebrafish is unknown. In that context, it is also noteworthy that human osteoarthritis, a condition due to degradation of cartilage in joints and increased subchondral bone formation, is characterized by modifications in the content of cartilage and bone ECM proteins [42].

Here, to further increase our understanding of skeletal formation using the zebrafish as a model system, we first investigated the early zebrafish osteoblast specific transcriptome by isolating *sp7*-expressing cells using the *Tg(sp7:sp7-GFP)* reporter line described earlier [43]. Based on their *sp7* expression, we identified two distinct subpopulations of osteoblasts that differentially express two ECM protein genes, *col10a1a* and *fbln1*. We therefore generated mutant zebrafish lines for these two genes to determine their previously unknown function in zebrafish bone development.

## 2. Materials and Methods

### 2.1. Fish and Embryo Maintenance

Zebrafish (*Danio rerio*) were reared in a recirculating system from Techniplast (Buguggiate, Italy) at a maximal density of 7 fish/L. The water characteristics were as follows: pH = 7.4, conductivity = 50 mS/m, temperature = 28 °C. The light cycle was controlled (14 h light, 10 h dark). Fish were fed twice daily with dry food (ZM fish food^®^, Zebrafish Management Ltd., Winchester, UK) with size adapted to their age, and once daily with fresh nauplii from *Artemia salina* (ZM fish food^®^). Larvae aged less than 14 days were also fed twice daily with a live paramecia culture. Wild type zebrafish from the AB strain and mutant lines were used. The *Tg(sp7:sp7-GFP)* transgenic line has been generated in-house as described earlier (ulg071 Tg) [43].

In general, two males and two females were used for breeding in the morning, eggs were collected and raised in E3 (5 mM NaCl, 0.17 mM KCl, 0.33 mM CaCl_2_, 0.33 mM MgSO_4_, 0.00001% methylene blue).

### 2.2. Dissociation of 4 Days Post Fertilization (dpf) Larvae to Obtain Osteoblasts and Preparation for FACS Sorting

A *Tg(sp7:sp7-GFP)* (ulg071 Tg) heterozygous transgenic parent was outcrossed with a Wild type (WT) parent to obtain a clutch of transgenic and non-transgenic offspring. At 3 dpf, the transgenic, fluorescent larvae were separated from their non-transgenic siblings and raised separately in two different plates. At 4 dpf, around 100–150 larvae were euthanized by adding 0.048% (*w*/*v*) of MS-222 (Ethyl 3-aminobenzoate methane sulfonate; Merck, Overijse, Belgium), and transferred to “gentle MACS™ C” tubes (Miltenyi Biotec, Leiden, The Netherlands) in an excess of E3 medium [44]. The supernatant was removed and replaced with 1–1.5 mL of de-yolk [44] buffer. The yolk was removed by vigorously pipetting up and down several times for 10 min and the supernatant discarded. The larvae were washed twice with Gibco^TM^ HBSS-buffer (without Ca^+2^ and Mg^+2^ ions and phenol red free) (Thermo Fisher Scientific, Merelbeke, Belgium) and the supernatant was discarded. The larvae were resuspended in 1.5 mL digestion buffer (1x Gibco^TM^ HBSS-, 10 mM HEPES (0.5%), 2 mM EDTA, TrypLe^TM^ Select 1x (Thermo Fisher Scientific, Merelbeke, Belgium), Proteinase K 0.2 µg/µL (Thermo Fisher Scientific, Merelbeke, Belgium), Collagenase-2 0.2 mg/mL (Worthington Biolabs, Lakewood, NJ, USA) at 28 °C and dissociation was performed running the protocols m_brain_01 and m_brain_03 (according to the manufacturer’s instructions in the Neural Tissue Dissociation Kit (T)) once on a gentleMACS™ Dissociator machine (Miltenyi Biotec, Leiden, The Netherlands), followed by three runs of m_brain_03, 2 runs of protocol m_brain_02, five runs of m_brain_03, each run separated by continuous shaking in the water bath at 28 °C for 5 min. Then, m_brain_01 protocol was run before adding 30 μL of Enzyme A as supplied by Miltenyi Biotec (Neural Tissue Dissociation Kit (T)) followed by 4 additional runs of the m_brain_03 protocol, separated by 5 min incubation at 28 °C with continuous shaking and the tube transferred on ice. The samples were centrifuged at 4 °C at 300× *g* for 10 min and the supernatant discarded as gently as possible to avoid cell resuspension. The pellet was very quickly resuspended in 1 mL HBSS^−^ buffer and filtered through mesh to eliminate cell clumps and aggregates. The filtered cells were collected in polypropylene tubes and viability dye 10% *v*/*v* propidium iodide (Fisher Scientific, Merelbeek, Belgium) was added to distinguish live cells from dead cells/cell debris while sorting. It is very important to ensure that the steps following centrifugation are carried out by placing the tubes on ice. The cells were then brought to the FACS Aria III sorter (BD Biosciences, Erembodegem, Belgium) and the green, fluorescent cells were sorted into a 1.5 mL Eppendorf tube containing 500 µL of PBS buffer with RNAse inhibitor (Promega, Leiden, The Netherlands) and 1% bovine serum albumin (BSA, Sigma-Aldrich/Merck, Overijse, Belgium). Note that a preliminary run was performed using the non-transgenic siblings to ensure that the GFP-positive cells were truly due to transgene expression, not to autofluorescence.

### 2.3. mRNA Sequencing

Cells coming from cell sorting (about 30,000–50,000 cells) were immediately lysed in 0.5% Triton X-100 containing 2 U/µL RNAse inhibitor (Promega, Leiden, The Netherlands) and stored at −80 °C. cDNAs were prepared from these lysed cells according to the SMART-Seq 2.0 protocol (Illumina, San Diego, CA, USA) for low input RNA sequencing, while libraries were prepared using the Nextera XT DNA library kit [45]. Only high-quality libraries were kept for sequencing (5 for the P1 and 3 for the P2 subpopulations). For the whole larvae mRNAs extracted from WT and *fbln1^−/−^* mutants, the cDNA libraries were generated from 100 to 500 ng of extracted total RNA using the Illumina Truseq mRNA stranded kit (Illumina, San Diego, CA, USA) according to the manufacturer’s instructions. All cDNA libraries were sequenced on a NovaSeq sequencing system (Illumina, San Diego, CA, USA), in 1 × 100 bp (single end). Approximatively 20–25 M reads were sequenced per sample. The sequencing reads were processed through the Nf-core rnaseq pipeline 3.0 [46] with default parameters and using the zebrafish reference genome (GRCz11) and the annotation set from Ensembl release 103 (www.ensembl.org; accessed 1 May 2020). Differential gene expression analysis was performed using the DESeq2 pipeline [47]. Pathway and biological function enrichment analysis was performed using the WEB-based “Gene SeT AnaLysis Toolkit” (http://www.webgestalt.org; accessed on 10 November 2022) based on the integrated GO [48], KEGG [49], Panther, and WikiPathways databases (all accessed on 19 April 2023 via http://www.webgestalt.org). Two additional databases were generated using data from zfin (zfin.org (accessed on 24 August 2022) and based on phenotypes associated with gene mutations (Geno-Pheno) or on location of gene expression (Expression).

### 2.4. Generation of Mutant Lines

Mutant lines for *fbln1* (zfin Id: ulg075) and *col10a1a* (zfin Id: ulg076) were generated using the CRISPR/Cas9 method as previously described [50,51]. The guide RNAs have been introduced into the Alt-R^TM^ Cas9 system from Integrated DNA Technologies (IDT, Leuven, Belgium), the gRNA sequence used are, respectively: 5′-CCTGGTGGCCTTGACGGCTGCCC-3′ for *col10a1a*, and 5′-CACCAGATAGTCACGCCCGT-3′ for *fbln1*.

The Alt-R crRNA (gRNA for the gene of interest) and tracrRNA were resuspended in nuclease-free IDTE Buffer to a final concentration of 100 μM each. The two components were mixed according to the manufacturer’s instructions, heated to 95 °C for 5 min, and cooled to 22 °C (gRNA). The 10 μg/μL Cas9 protein was diluted to 0.5 μg/μL using Cas9 Buffer consisting of 20 mM HEPES, 150 mM KCl with pH = 7.5 in a final volume of 10 μL. Then, 3 μL of gRNA were mixed with 3 μL of the Cas9 solution, incubated at 37 °C for 10 min, cooled to 22 °C, and mixed with a 0.5 μL tracer dye (0.5 mg/mL, Rhodamine dextran (RD), Molecular Probes, Carlsbad, CA, USA). Microinjection of 2–3 nL per embryo was carried out on single cell stage (20–60 min post-fertilization) zebrafish embryos using an InjectMan micromanipulator (Eppendorf, Hamburg, Germany) assembled on a Leica M165 FC stereomicroscope.

DNA was isolated from whole larvae or fin clips from adults/juveniles at various stages of development in 50 mM NaOH by heating at 95 °C for 20 min. The solution was cooled down on ice for 10 min, neutralized by adding Tris-HCl 1 M, pH = 8.0, 1/5th the volume of NaOH, spun down using a desktop centrifuge for 2 min to recover the supernatant, and stored at 4 °C. Genomic fragments covering the targeted region were obtained by PCR using the above primers. The primers for PCR genotyping were, respectively, forward 5′-CAGATTTGACTTCAGAGAATGGA-3′, reverse 5′-AGAAACACAGCTTTTCCGAGAG-3’ for *col10a1a* and forward 5′-GTTGGGTCAGATGTGCTGTG-3′, reverse 5′-ATGAGTCTGACCGTGTGCTG-3′ for *fbln1*. The mutants were identified using Heteroduplex Mobility shift Analysis (HMA) by polyacrylamide gel electrophoresis; selected DNAs were further processed for Sanger sequencing to identify the exact position and extent of the mutation.

### 2.5. Genotyping and RNA Extraction of WT and Mutant Larvae

Homozygous mutants were obtained by crossing heterozygous parents carrying the desired mutation. Resulting larvae were first sacrificed, fixed in para-formaldehyde (PFA) 4% or stained, then photographed, and finally DNA was extracted from individual larvae to genotype them as described above. Only homozygous WT or mutants were then assigned to their photograph for phenotype determination.

For the RNA-Seq experiment on mutants, 10 dpf larvae were stored in RNA later (Fisher Scientific, Merelbeek, Belgium). Individual fish were decapitated, the heads were individually stored in a 96 well plate, while the body was used for DNA extraction and genotyping. Once the genotypes were known, the heads were recovered and pooled to constitute three independent batches of, respectively, 21 WT and mutant individuals. The RNA was extracted using the RNA mini extraction kit (Qiagen, Hilden, Germany) according to the manufacturer’s instructions. The RNA was treated with DNAseI (Qiagen, Hilden, Germany) to avoid DNA contamination. Quantity and quality of each extract was assessed by nanodrop spectrophotometer measurements. The pellets were further purified by lithium chloride precipitation, followed by pellet washing twice with 70% ethanol, resuspended in 51 µL of RNAse-free water, and stored at −80 °C. The integrity of total RNA extracts was assessed using the BioAnalyzer (Agilent, Santa Clara, CA, USA). RIN (RNA integrity number) scores were >9 for each sample.

### 2.6. Alizarin Red (AR) Staining

Larvae were sacrificed at 5 dpf and 10 dpf. The larvae were fixed in PFA 4% overnight at 4 °C and thereafter rinsed three times with Phosphate Buffered Saline/0.1% Tween (PBST) for 10 min. Bleaching was performed by adding 6 mL of H_2_O_2_ 3%/KOH 0.5% during 30 min for 5 dpf and 45 min for 10 dpf, respectively, followed by washing twice for 20 min with 1 mL 25% glycerol/0.1% KOH to remove bleaching solution. The larvae were stained with AR (Merck, Overijse, Belgium) at 0.05% in the dark for 30 min on low agitation. Rinsing and destaining was performed thrice at 50% glycerol/0.1% KOH for 30 min. The solution was replaced with a fresh solution of 50% glycerol/0.1% KOH and stored at 4 °C. The larvae were placed in lateral or ventral view onto glycerol (100%) for imaging. Images of stained larvae (n = 60–100 larvae) in three or more independent experiments were obtained on a stereomicroscope (Olympus model SZX10, Tokyo, Japan, cell B software version 3.4).

### 2.7. Alcian Blue (AB) Staining

Larvae were sacrificed by exposure to MS-222 (Ethyl 3-aminobenzoate methane sulfonate; Merck, Overijse, Belgium) (0.048% *w*/*v*) at 5 dpf and 10 dpf. The larvae were fixed in PFA 4% over night (ON) at 4 °C and thereafter rinsed three times with PBST for 10 min. The larvae were stained with 1 mL of alcian blue at 0.04% alcian blue (Sigma-Aldrich/Merck, Overijse, Belgium)/10 mM MgCl_2_/80% EtOH pH 7.5 O/N, on low agitation. Thorough rinsing was performed at least 7 to 8 times with 80% EtOH/10 mM MgCl_2_/water, on low agitation till excess of blue stain is washed and the washing solution appears clear. The larvae were washed with 50% EtOH pH 7.5 for 5 min and then with 25% EtOH pH 7.5 for 5 min. Bleaching was performed by adding 6 mL of H_2_O_2_ 3%/KOH 0.5% during 30 min for 5 dpf and 45 min for 10 dpf, respectively. Then, washing was performed twice for 20 min with 1 mL 25% glycerol/0.1% KOH to remove bleaching solution. Rinsing and destaining was performed three times at 50% glycerol/0.1% KOH for 30 min. The solution was replaced with a fresh solution of 50% glycerol/0.1% KOH and stored at 4 °C. The larvae were placed on the lateral side or ventral side onto glycerol (100%) for imaging. Images of stained larvae (n = 60–100 larvae) in three or more independent experiments were obtained on a stereomicroscope Olympus model SZX10, Tokyo, Japan, cell B software version 3.4), taking care to always use the same lighting and magnification conditions.

### 2.8. Image Analysis of Larvae Stained for Cartilage or Bone

Image analysis was performed on the pictures of larvae stained with alcian blue for cartilage or alizarin red for bone. According to [52], cartilage (alcian blue) images were analyzed by measuring the distances from anterior to the posterior end of the ethmoid plate (head length-hl), between the two hyosymplectics (d-hyo), between the articulations joining the Meckel’s cartilage to the palatoquadrate (d-art), and the angle formed by the two ceratohyals (a-cer); while the degree (absent, low, normal/intermediate, high) of bone mineralization (alizarin red) was visually estimated from images of the following bone elements [53]: maxillary (m), dentary (d), parasphenoid (p), entopterygoid (en), branchiostegal rays 1 and 2 (br1/br2), opercle (o), ceratohyal (ch), hyomandibular (hm), vertebral bodies (vb).

### 2.9. Micro-Computed Tomography Scanning (µCT) and Analysis

WT and their respective mutant siblings were grown in the same tank at identical zebrafish density to minimize variability. The zebrafish were sacrificed, their standard length documented, then fixed for 14–16 h at 4 °C in 4% (*w*/*v*) PFA and prepared for µCT imaging. The individual zebrafish were kept hydrated in a sponge covering and placed in a sample holder during µCT acquisitions (SKYSCAN 1272 scanner (Bruker Corporation, Kontick, Belgium)). Whole body scans were acquired at 70 kV and 100 µA with a 0.5 mm aluminum filter and at an isotropic voxel size of 21 µm. For high-resolution scans and quantitative analysis of the first precaudal vertebrae, zebrafish were scanned at 70 kV and 100 µA with a 0.5 mm aluminum filter at an isotropic voxel size of 7 µm. For all samples, the beam hardening correction parameter was held constant, while ring-artifact correction is sample-dependent, so it was adapted and applied to each sample. No smoothing was applied during image reconstruction (NRecon, Bruker). Images with 7 µm voxel size were manually segmented using pmod version 4.0 (PMOD Technologies, Zurich, Switzerland) to extract precaudal vertebrae 6–8 and both vertebral thickness and vertebral length were determined.

Further analysis of the 21 µm images was performed using the FishCuT Software version 1.2 [54,55]. Briefly, FishCut is a matlab toolbox [54] designed to analyze microCT image of zebrafish and extract morphological and densitometric quantitative information of zebrafish for 25 vertebrae per individual in the case of *fbln1^−/−^* mutants, whereas for *col10a1a^−/−^* mutants, only 23 vertebrae were analyzed due to the fusion of the caudal fin vertebrae. Since FishCut was initially developed on images obtained with a vivaCT40 (Scanco Medical, Wangen-Brüttisellen, Switzerland), we first adapted the parameters (intercept and slope) that should be used in the tissue mineral density (TMD) conversion formula [54]. These parameters were estimated from the calibration scan performed on the same day of the data acquisition. The following combinatorial measures were considered and quantified for each vertebra: centrum surface area (Cent.SA), centrum thickness (Cent.Th), centrum tissue mineral density (Cent.TMD), centrum length (Cent.Le), haemal arch surface area (Haem.SA), haemal arch thickness (Haem.Th), haemal arch tissue mineral density (Haem.TMD), neural arch surface area (Neur.SA), neural arch thickness (Neur.Th), neural arch tissue mineral density (Neur.TMD), vertebral surface area (Vert.SA), vertebral thickness (Vert.Th), vertebral tissue mineral density (Vert.TMD), and were measured. Vertebral measures (Vert) represent the total vertebral body, with all three elements (centrum, haemal arch, neural arch) combined.

### 2.10. Statistical Analysis

Statistics were performed using GraphPad Prism9 software (v. 9.4.1). An unpaired *t*-test was used for comparing distances (d-hyo, hl, and d-art) and angle (a-cer) for cartilage elements in larvae, while for vertebral thickness and length in adults an ordinary one-way ANOVA was used. For comparing the degree of mineralization, we used a Chi-square test on contingency table between WT and mutants. Multivariate analysis (Multiple linear regression analysis) was used for statistical analysis of the FishCut output data. Comparison of left and right opercular areas were performed using Sidak’s multiple comparisons test of mixed effect analysis. All the values are expressed as mean ± SEM and statistical significance was set at *p* < 0.05.

## 3. Results

### 3.1. Analysis of sp7-Expressing Cells from Transgenic Zebrafish Larvae Reveals the Presence of Two Distinct Osteoblast Populations at 4 dpf

To analyze the transcriptome of living osteoblasts isolated from developing larvae, we employed the transgenic line *Tg(sp7:sp7-GFP)* (ulg071 Tg) that carries the GFP reporter cDNA inserted into the endogenous, osteoblast-specific *sp7*(*osterix*) gene [43]. Transgenic larvae were dissociated at 4 dpf and the cells were analyzed using Fluorescence Activated Cell Sorting (FACS). Interestingly, when looking at the fluorescence distribution of individual cells, we observed two clear subpopulations based on their GFP fluorescent intensities, referred to as P1 (weakly positive for GFP) and P2 (strongly positive for GFP), respectively (Figure 1A). This fluorescent signal was not observed in cells obtained from non-transgenic siblings, proving that it originated from truly Sp7-GFP positive cells and not from some autofluorescent cells (not shown). Whole transcriptome RNA sequencing was performed to compare the transcriptomes of these two subpopulations (P1 and P2). In addition, we also compared each subpopulation’s transcriptome to that of a 4 dpf whole larvae gene expression data set (“All”) from a public database (https://www.ebi.ac.uk/ena/browser/view/PRJEB7244 (accessed on 1 May 2020).

Analysis of the Differentially Expressed Genes (DEGs) (*p* < 0.001, log(fold-change) > 1.6) lists revealed, respectively, 4308 DEGs in P1 and 4531 DEGs in P2 relative to the “All” population (“AllvsP1” and “AllvsP2”), but only 966 DEGs were observed between P1 and P2 subpopulations (“P1vsP2”) (Figure 1B and Appendix A). A Venn diagram analysis showed that a vast majority of genes (3648) were common to the “AllvsP1” and “AllvsP2” (Figure 1C), which also encompassed a majority of those differentially expressed in the subpopulations “P1vsP2”. Principal Component Analysis revealed that the two subpopulations P1 and P2 were very distinct from the general “All” cell population (PC1; 89% of the variance); however, the P1 and P2 cells clearly clustered separately (PC2; 4% of the variance), indicating that they may be distinct subpopulations of osteoblasts (Figure 1D). We then analyzed the different DEG lists for enrichment in various databases (GO terms, KEGG, Panther, Reactome, and Wikipathways) to identify biological functions, cellular components, or pathways that may be affected. GSEA analysis revealed that, relative to “All”, in the P1 population (“AllvsP1”) immune response and cardiovascular development were the top biological processes that were up-regulated, while in the P2 subpopulation (“AllvsP2”), ossification and biomineral tissue development were significantly induced (Figure 1E and Appendix A). Also, “extracellular region part” and “collagen trimer” were identified as the major cellular components.

Finally, comparison of the P1 and P2 transcriptomes clearly identified “biomineral tissue development”, “ossification”, and “Endochondral ossification” as significantly up-regulated in P2, as well as the signaling pathways initiated by TGFbeta, BMP, Wnt, and MAPkinases. GSEA analysis using databases for known phenotypes upon mutation (Pheno-Geno in Appendix A) or expression domains of specific genes in zebrafish (Expression in Appendix A) revealed a significant enrichment in genes expressed in and affecting skeletal elements for genes up-regulated in the P2 subpopulation, relative to both P1 and “All”. Cardiovascular genes were also up-regulated in P2, while both subpopulations revealed down-regulation of genes involved and expressed in neural development relative to “All”.

Taken together, a picture emerges of a P1 population that is already engaged into differentiation into skeletal cells, while the P2 population appears as resolutely committed to the osteoblast fate with many specific genes being strongly upregulated, such as *runx2b*, *col10a1a*, or *spp1*. Therefore, we will in the following analysis consider the P1 population as osteoblast-like cells clearly distinct from the general cells of the whole larva (“All”), and possibly a precursor to the more mature P2 osteoblast population.

Different patterns emerged for individual genes when looking at their changes in expression in the different cell populations, as illustrated for selected genes displayed in a clustered heat map based on the change in the number of reads relative to the “All” population (Figure 2A). Genes unaffected or slightly upregulated in P1, but strongly upregulated in P2 would be known osteoblast markers such as *sp7* (and the transgene GFP, as expected from the FACS sorting method), but also *bmp2a*, *col10a1a*, *panx3*, and *spp1* (Figure 2B). Others were increased in P1, and remained high in P2 (*fbln1*, *col1a1*, *col2a2a*, *col2a2b*). The last group of genes show increased expression in P1 relative to “All”, and a decrease in P2 (*lrp2a*, *omd*, *stcl1*). These observations indicate the involvement of a complex pattern of gene regulations comparing the general larvae cell population “All” to the P1 and P2 subpopulations.

To gain further insight into the mechanisms involved in these differentiation processes, we decided to overlay the differential expression data between P1 relative to “All”, and those between P1 and P2, to specific process networks. First, we collected the genes with annotations “Mineralization” or “Ossification” to construct a network of genes involved in the main function of osteoblasts (Figure 3) [56]. Genes upregulated in both P1 and P2 include the bone-related *sp7* and *vdra* transcription factor genes and the ECM protein genes (*spp1*, *bglapl*, *col1a1*, *mgp*). Genes for enzymes controlling extracellular phosphate concentration to ensure mineralization *(entpd5a*, *alpl*, *phospho1*) were strongly induced in both populations, as were those for some extracellular peptidases (*mmp9*, *mmp14a,* and *b*) and some cell membrane proteins such as *panx3* and *tmem119a*. Activation of the Hedgehog pathway is clearly indicated by the induction of the *ptch1* and *ptch2* genes, while the Bmp pathway appears to be activated mainly through *bmp3*, *bmp1a*, and *ltbp1*. The Wnt pathway is mainly activated by *wnt5b* in both P1 and P2, while the *wnt9a* gene is mainly induced in P2, and the *wls* and *lrp5* genes in P1. Thus, complex gene expression patterns emerge distinguishing the two osteoblast populations P1 and P2.

We then looked more specifically at the BMP signaling network (Appendix A). It appears very clearly that the genes for ligands Bmp2a, Bmp3, and Bmper were strongly upregulated in both P1 and P2, while *bmp4*, *bmp6*, and *bmp7a* were more strongly induced in P1, or *bmp1a* and *bmp2b* only in P2. Other genes are only weakly affected (*bmp1b*) or follow a decrease-increase pattern (*bmp15*). Note that Bmp1 proteins are not *bona fide* BMP ligands but are rather involved in the collagen synthesis process through their peptidase activity. Considering BMP receptors, we observe that the *bmpr1aa*, *bmpr1ba*, *bmpr1bb*, and *acvrl1* genes are strongly induced mainly in P1, while *bmpr2a* and *acvr2ba* are induced both in P1 and P2. Genes *smad4a*, *smad4b*, *smad6b*, and *smad9*, coding for the Bmp signal transducing transcription factors, are up-regulated in both subpopulations. Down-regulated genes in P2, relative to P1, are the transcription factors of the Gata family, involved in hematopoiesis, cardiac and vascular development, as well as genes coding for extracellular inhibitors of BMP signaling, such as follistatins (*fsta*, *fstb*, and *fstl1a*). Other BMP antagonists are mainly upregulated in P1 and not changed in P2 (*grem1b*, *grem2a*), while *bambia* and *bambib* are strongly upregulated in both subpopulations, possibly reflecting their role in enabling Wnt signaling [57,58].

Analysis of the WNT signaling pathway leads to similar observations (Appendix A). The *wnt1*, *wnt5b*, and *wnt10a* genes are upregulated in both P1 and P2, while *wnt5a*, *wnt11*, *wnt11f2*, and *wnt7bb* are strongly upregulated in P1 and those for *wnt3* and *wnt6b* are downregulated. Some of the WNT ligand genes are up-regulated in P1, and down-regulated in P2 (*wnt7aa*, *wnt7ab*, *wnt5a*, and *wnt4b*). In terms of receptors, *fzd1*, *fzd2*, *fzb8b*, *lrp4*, *lrp5*, and *lrp6* are mainly up-regulated, as well as the downstream mediators such as *axin2*, disheveled genes *dvl1a* and *dvl1b*, and the beta catenins *ctnnb1* and *ctnnb2*. WNT pathway target transcription factor genes *jun* and *tcf7* are strongly up-regulated.

Taken together, a picture emerges where, compared to the general cell population “All”, two distinct subpopulations of osteoblasts are present in 4 dpf zebrafish larvae based on their *sp7* (or the transgene GFP) expression. Both subpopulations present clear features of skeletal cell gene expression, with the P2 population more clearly identified as osteoblasts. However, the different components: transcription factors, ECM proteins, signaling ligands, receptors, and downstream effectors present a complex pattern of changes in expression in the two subpopulations, probably reflecting the requirement for precise coordination and regulation of the various players involved.

The ECM protein gene *col10a1a* attracted our special attention at this stage. First, in contrast to its specific expression in hypertrophic chondrocytes as is generally accepted in mammals, it is also expressed in osteoblasts in zebrafish [18,25,29]. Moreover, our analysis of the osteoblast transcriptome at 4 dpf revealed *col10a1a* as one of the most highly upregulated genes in the P2 osteoblast population relative to P1 (Appendix A, Figure 2), while it was slightly down-regulated in P1 relative to “All” (Figure 3, log(fold-change) = −1.34, *p*-value = 4.7 × 10^−7^). These considerations prompted us to investigate its function in zebrafish, which was unknown.

Another ECM protein whose function was unknown in zebrafish is Fibulin1, although its involvement in skeletal development was already shown in a mouse model [37]. The *fbln1* gene, different from *col10a1a*, displayed increased expression in the P1 subpopulation relative to “All”, and a further increase in the P2 population (Figure 2A,B), thus following the expression pattern of *sp7*. We thus decided to also investigate the function of this gene in zebrafish skeletal development.

### 3.2. Analysis of col10a1a^−/−^ Zebrafish Mutants

To elucidate the role of *col10a1a* in zebrafish skeletal development, we generated a mutant line (ulg076) disrupting the *col10a1a* coding region with a 34-nucleotide insertion and we compared the effects of this mutation in the developing larvae and in adults. No difference was observed in the standard length of WT compared to *col10a1a^−/−^* siblings at 10 dpf, respectively (*p* value = 0.97). Staining of the larvae at 5 dpf with Alcian blue to elucidate the effects of the mutation on cranial cartilage (Figure 4A) revealed a significant reduction of the chondrocranium in the *col10a1a* mutants, as indicated by the smaller distance between the two hyosymplectics (d-hyo) and the head length (hl) in mutants compared to WT (Figure 4B). Alizarin Red staining at 10 dpf revealed an overall decreased mineralization in *col10a1a* mutants for elements of the cranial skeletal such as m, d, en, p and br1 in comparison to WT controls (Figure 4C,D). These observations are consistent with those made using another mutant carrying a 7-nucleotide deletion at the same location (not shown).

The mutant larvae and their WT siblings were grown throughout adulthood and the one-year-old fish were subjected to µCT analysis. No significant difference in the standard length was observed between WT and *col10a1a* mutants (n = 6, *p* = 0.259). No major deformities were detected in the head or vertebral column; however, a projected image of a µCT scan revealed a decreased mineralization in *col10a1a^−/−^* fish relative to WT (Figure 5A), and fusion of vertebral bodies at the tail fin was detected (Figure 5A,F). The µCT images were analyzed using two different approaches. First, three precaudal vertebral bodies (number 6–8) were selected, as shown in (Figure 5B), and morphometric measurements of vertebral thickness (at equivalent positions) and vertebral length were carried out as illustrated in the three different planar sectional view (Figure 5C). This analysis (n = 4 fish/group) revealed a significantly decreased vertebral length (*p* < 0.05) and vertebral thickness in all three vertebrae in *col10a1a^−/−^* adult zebrafish compared to WT (Figure 5D). An additional analysis was performed by quantifying combinatorial measures over the entire vertebral column (Figure 5E). The TMD of the vertebrae and centra, as well as the haemal and neural arches (not shown) (n = 6 individuals/group and 23 vertebrae/individual) was significantly decreased in the *col10a1a^−/−^* mutants compared to WT controls (*p* < 0.05) (Figure 5E), while only surface area were affected in centra and neural arches (Cent.SA, Neur.SA). A closer look at the µCT of the tail fin vertebrae revealed a complex vertebral fusion in four out of six *col10a1a^−/−^* animals (Figure 5F).

### 3.3. Analysis of the fbln1^−/−^ Mutant Line

To elucidate the role of *fbln1* in skeletal development, we generated a mutant line presenting a 16-nucleotide deletion in the *fibulin1* coding region (ulg075) and performed phenotypic analysis by staining for cartilage using AB and for bone using AR staining at developmental stages 5 dpf and 10 dpf. No significant differences were observed in standard length for *fbln1^−/−^* mutants at 5 dpf (*p* value = 0.671) or 10 dpf (*p* value = 0.857) compared to WT. Looking at the 5 dpf chondrocranium (Figure 6A), we observed that the *fbln1^−/−^* mutants exhibit significant reduction in the ceratohyal angle (a-cer) (*p*-value = 0.0154), while no significant effects were observed in the other measures (d-hyo, d-art, hl) (Figure 6B). Looking at the calcified bone matrix (Figure 6C–F), the level of mineralization for skeletal elements, such as en, br1, p, and vb, was significantly increased in mutants at 5 dpf and 10 dpf (Figure 6D,F).

No lethality or no major physical defect were observed in the on-growing *fbln1^−/−^* mutants, thus we let them grow alongside their WT siblings until one year of age to perform µCT analysis. High mineralization was easily visible in *fbln1^−/−^* zebrafish in a whole body projected image of the µCT scans compared to WT (Figure 7A). Using the pmod software (version 4.0), we segmented the precaudal vertebral centra of the precaudal vertebrae 6–8 (Figure 7B). Morphometric measurements, as illustrated in the three different planar sectional views (Figure 8C), revealed (n = 4 fish/group) a significantly increased vertebral length (*p* < 0.01) and vertebral thickness (*p* < 0.0001) in *fbln1^−/−^* adults relative to WT (Figure 7D). Using the FishCuT software version 1.2 to perform combinatorial measurements on all individual 25 vertebral bodies/fish [54], we observed a significantly increased volume (Cent.Vol), surface area, (Cent.SA), thickness (Cent.Th), and TMD (Cent.TMD) for the vertebral centra in *fbln1^−/−^* as compared to the WT controls (n = 7 individuals/group, *p* < 0.05) (Figure 7E). In addition, whole vertebra TMD (Vert.TMD) was increased, while overall neural arch thickness (Neur.Th) was not affected.

To complete our analyses of the *fbln1^−/−^* line, whole larvae mRNA was extracted from WT and homozygous mutant larvae at 10 dpf to compare their expression level by whole genome sequencing. 2511 DEGs (*p*-value < 0.05, log(fold-change) > 0.5) were observed (2214 upregulated, 297 downregulated in the mutant) (Figure 8A,B and Appendix A). Among the most strongly upregulated genes were the *col10a1a*, *col1a1a*, *col11a1a*, and the *col2a1a* genes, but also many bone-related genes such as *entpd5*, *enpp1*, *sp7*, and *spp1* (Figure 8A). Among the small number of downregulated genes, we spotted *runx2b* and *alpl*, both markers for osteoblast differentiation, which were minimally affected. The *fbln1* mRNA was decreased in the mutant, indicating that the mutation caused some degree of RNA degradation. Functional annotation using GSEA revealed that mainly one biological process was upregulated: collagen biosynthesis, trimerization, and endochondral ossification (Figure 8B). Interestingly, all three signaling pathways for Bmp, Wnt, and Fgf ligands were identified as slightly downregulated. We constructed a network around collagen biosynthesis which revealed the consistent upregulation of all the collagen genes, but also some genes coding for integrins *(itgb1a*), matrilins, fibronectin, or genes coding for enzymes involved in collagen maturation (*plod1a*, *plod2*, *p3h1*, *p3h3*) (Figure 8C). Taken together, these results correlate with an increase of bone matrix deposition and mineralization, although some markers for osteoblast differentiation were not significantly affected.

### 3.4. The fbln1^−/−^ Mutant Lacks an Opercle Specifically on the Right Side

Another unexpected phenotype that we observed was that 75% of 3-month-old *fbln1^−/−^* mutants were missing an opercle on the right side (9 out of 12 individuals), compared to none in the WT controls. Simple observation of one-year-old adult homozygous *fbln1^−/−^* zebrafish clearly revealed this missing opercle, always on the right side compared to the contralateral left side or the WT (Figure 9A). The µCT scans of *fbln1^−/−^* adult zebrafish heads confirmed this observation (Figure 9B). In addition, closer examination of these µCT scans of *fbln1^−/−^* adults revealed a thickening of the cavity walls and of the subopercular bone at the site of the missing opercle (Figure 9C). The surprising asymmetry in opercle development in the adults prompted us to find out when it arises, and to measure the opercle area on alizarin red stained 10 dpf larvae, this time comparing the left (L) and right (R) opercle in mutants and WT (Figure 9D). The *fbln1^−/−^* mutants exhibited a trend (*p* = 0.121) to a smaller opercle on the right side compared to the left one, which is not observed in WT siblings. This unique phenotype is reminiscent of the condition previously observed in *ace^ti282a/fgf8^* heterozygote zebrafish due to *fgf8* haploinsufficiency [59]. These *fgf8^ti282a/+^* presented an asymmetric jaw and various size reductions of the opercle always on the right side, often displaying fusion of the opercle with the branchiostegal rays1 and 2. *fgf8* expression was observed in distinct regions of the opercle, jaws, and in the cranial sutures [59]. Thus, our results suggest a probable role of Fbln1 in modifying Fgf8 signaling specifically regulating opercle development. To further investigate this hypothesis, we analyzed the changes in gene expression in the *fbln1^−/−^* mutants specifically in the context of FGF signaling (Figure 9E). Interestingly, we observe a slight downregulation of the *fgf8a* and *fgf8b* genes, along with induction of FGF receptor genes *fgfrl1a* and *fgfrl1b*, and several MAPkinase genes. Looking at known marker genes and specific targets for Fgf8 signaling [60], we observe that *etv5a* is slightly down-regulated (log(fold-change) = −0.32, *p* = 0.016).

## 4. Discussion

Omics technologies have been used in the past to reveal the entire transcriptome of osteoblasts from human bone marrow progenitor cells [61], differentiating primary fibroblasts [62], or primary mouse bone marrow stem cells [63,64,65] by micro-array analysis. More recently, deep sequencing was applied to analyze the transcriptome of human osteoblasts from osteoarthritic patients [66] or from mouse tibias [67]. In all these studies, the cells were isolated from a specific location (often calvarial bone marrow) in adult individuals and kept in culture before submitting them to a differentiating treatment. Some recent studies investigated primary osteoblasts directly after surgery, such as comparing primary human osteoblasts to osteoblastomas [68] or mouse primary calvarial osteoblasts [69]. Interestingly, one such study revealed that cells isolated from rat skull or ulnar bones presented distinct transcriptomes [70]. Recently, the single cell transcriptome of two-month-old zebrafish tail muscle tissue revealed the presence of osteoblast-type cells [71]. These studies confirmed the presence of crucial transcription factors, such as SOX9, RUNX2, and SP7 in differentiating osteoblasts, but they all relied on cells obtained from mature tissues in adults.

We decided to take advantage of the quite unique opportunity given by the zebrafish model to obtain, easily and without dissection, developmentally early osteoblasts from 4 dpf larvae. This was achieved by isolating fluorescent cells from transgenic larvae carrying the *GFP* reporter cDNA inserted into the endogenous *sp7* gene. Our assumption that this was the best way to drive GFP expression into developing osteoblasts is verified by the RNA-Seq analysis, as the highly fluorescent cells (subpopulation P2) preferentially express bone-related genes (such as *sp7* itself, but also *runx2b*, *spp1*, *entpd5a, alpl*, *col10a1a*), while the functional annotations specifically point to induction of ossification, ECM formation, and mineralization. This identifies subpopulation P2 as mature osteoblasts. The presence of a larger, weakly fluorescent population (subpopulation P1) came as a surprise. These cells are distinct from the P2 cells; however, many of the genes involved in bone development are also expressed, albeit at lower levels (*sp7*, *bmp2a*, *spp1*, *fbln1*) (Figure 2 and Appendix A). Other genes, such as *col1a1*, *col2a2a*, and *col2a2b* are equally expressed in P1 and P2, while still others are downregulated in P2 relative to P1 (*hoxb9a*, *omd*, *stc1l*). Thus, while it is tempting to speculate that population P1 would constitute a discrete osteoblast precursor population due to its very similar transcriptome to P2 (Figure 1C,D), it is at present unclear whether the P1 and P2 subpopulations can be placed in a continuous differentiation lineage or whether they represent two independent lines. One possibility would be that the P2 osteoblasts are related to intramembranous ossification, as at 4 dpf mainly this type of bone elements is mineralizing. The P1 population would correspond to the higher number of precursor cells. These precursor cells do not (or not yet) express late marker genes, such as *spp1* or *col10a1a*. This is also consistent with previous reports showing *col10a1a* expression nearly exclusively in bones at 4 dpf [23,34], while its expression in chondrocytes is observed at 6 dpf [72].

Further comparison of the P1 and P2 subpopulations revealed various patterns of gene expression (Figure 3, Appendix A), illustrating the complex regulatory events occurring in these cells. Two other genes caught our attention: the *stanniocalcin 1-like* (*stc1l*) gene and the *osteocrin* (*ostn*) gene, which both were strongly upregulated in P1, but downregulated in P2 (Figure 2B and Figure 3). These genes play a role in calcium homeostasis [73] and skeletal development [74,75], respectively, and both are expressed specifically, but not exclusively in the corpuscles of Stannius. Whether this means that the P1 population contains corpuscule of Stannius cells, which would express low amounts of *sp7*, or whether these genes are also expressed transiently in osteoblast precursor cells will need to be further investigated.

Further insight into the function of the two subpopulations may be gained from the phenotypes of the mutants that we analyzed. The *col10a1a* gene is interesting in this respect, as it is downregulated in the P1, but dramatically upregulated in the P2 subpopulation. Furthermore, its mammalian homolog is considered not to be expressed in osteoblasts. This expression pattern is consistent with the report that the Sp7 transcription factor directly regulates the *col10a1a* gene during zebrafish development [18]; however, it appears that only high *sp7* expression leads to high *col10a1a* expression. The other ECM protein gene that drew our attention is the *fibulin1 (fbln1)* gene, which is upregulated in both P1 and P2 subpopulations. *col10a1a* and *fbln1* mutant analysis revealed opposing effects with, respectively, a decreased or increased mineralization in both larval and adult stages. These phenotypes are consistent with a role for the *col10a1a* gene mainly in the P2 mature osteoblasts, while *fbln1* is already required in the P1 precursors, possibly by facilitating proliferation or recruitment to the osteogenic lineage. Such a role would be consistent with the observed increase in expression of mature osteoblast genes in the *fbln1^−/−^* animals. One of these genes is actually *col10a1a*, which is required for bone mineralization as shown by the *col10a1a* mutant.

The decreased mineralization we observed in the *col10a1a* mutants is coherent with the observations made in mice and humans. It is also consistent with the high expression of this gene in osteoblasts, which also extends to the axial skeleton at 6 dpf and later in the entire vertebral column, in medaka [24], and in zebrafish [76]. Interestingly, although *col10a1a* is a direct transcriptional target of the Sp7 transcription factor, we did not observe the dramatic defects in opercula, tail fin, and craniofacial development that were described in *sp7^−/−^* mutants [18], indicating that Col10a1a is only one of the factors regulated by Sp7 to be required for correct osteogenesis. We only observed a vertebral fusion at the tail end of the vertebral column in the *col10a1a* mutants.

In the *fbln1^−/−^* mutants, the increased mineralization is consistent with the observed *fbln1^−/−^* transcriptomic profile, showing a significant increase in several of the genes for collagens and for enzymes involved in collagen biosynthesis and maturation. Interestingly, many of the genes upregulated in *fbln1^−/−^* mutants at 10 dpf are also among those that are upregulated in the P1 and P2 osteoblast subpopulations at 4 dpf, such as *col10a1a*, *entpd5a*, *enpp1*, and *spp1*. Some of these genes are also target genes for the transcription factor Sp7 [59], whose expression is significantly induced in the *fbln1^−/−^* mutant (Figure 8A). This is a sharp difference to the mouse *Fbln1* KO, where *Sp7* expression was clearly reduced [37], possibly explaining the divergent effect of Fibulin1 depletion in zebrafish and mouse. Increased Sp7 expression may be one of the driving mechanisms for the increased bone matrix deposition and mineralization in *fbln1* mutants, even though some of the osteoblast differentiation marker genes like *runx2b* and *alpl* are downregulated.

In contrast to the findings in *Fbln1* KO mice, the survival rate of *fbln1^−/−^* zebrafish is normal until adulthood and without any malformations or defects that could impact its development and growth. A possible explanation for this discrepancy may be given by the fact that the *fbln1* gene belongs to a family of eight ECM proteins [35], whose increased (possibly ectopic) expression may lead to the recently described phenomenon of “transcriptional adaptation” [77]. In zebrafish, members of this family such as *fbln7 or fbln8* are expressed in skeletal structures [78], and *hmcn2* is co-expressed with *fbln1* in the fin mesenchymal cells and developing somites [40]. Increased expression of FBLN2 was indeed shown to rescue the function of FBLN1 in the placenta of KO mice [38,79]. Our analysis of the *fbln1^−/−^* mutant transcriptome, compared to WT, did reveal a downregulation of the *fbln1* gene, together with an upregulation of *fbln2*, *fbln5*, *hmcn1*, *hmcn2* at 10 dpf. Thus, although the general function of these genes remains largely unknown in zebrafish, we cannot rule out the possibility that one of them may rescue the lethal phenotype in our *fbln1^−/−^* mutant zebrafish, as described in mouse.

Other, more morphological effects probably result from perturbations of morphogenetic signaling pathways, as Fibulins are known to interact with signaling molecules such as BMPs, WNTs, or FGFs [35]. The most striking defect was obviously the missing opercle on the right side, as detected in 75% of *fbln1^−/−^* mutants (Figure 9), for which a trend was already detectable in 10 dpf mutant larvae (Figure 9D). Closer inspection revealed a thickening of the opercular cavity walls and of the subopercular bone at the location of the missing opercle (Figure 9B), suggesting that fusion of the developing opercle occurred in these animals as was described in haploinsufficient *fgf8^ti282a/+^* mutants [59]. Transcriptome analysis of the *fbln1*^−/−^ mutant was consistent with a downregulation of the FGF8 signaling pathway. The role of Fgf8 in zebrafish left–right asymmetry was previously shown [80], while the Fgf8 dosage was also shown to influence craniofacial shape and symmetry in mice [81]. Furthermore, tight binding has been shown between mouse FBLN1 and FGF8, while downregulation of FBLN1 inhibits FGF8 expression [82]. Taken together, these observations suggest that *fbln1* mutation may lead to downregulation of FGF8 signaling, resulting in absence of the opercle on the right side.

## 5. Conclusions

Taken together, we show that a population of *sp7*-expressing osteoblasts isolated from 4 dpf zebrafish larvae could be separated in two subpopulations, each one characterized by a specific expression pattern of bone-promoting genes. Pathway analysis revealed a complex pattern of signaling pathway components, transcription factors, and ECM protein genes that characterize each of the subpopulations. Investigation of mutant zebrafish for two genes encoding ECM proteins revealed that both *col10a1a* and *fbln1* play important roles in maintaining skeletal integrity, interestingly with opposite effects. Our results point to a central role for the transcription factor Sp7, activating expression of the *col10a1a* gene in regulating bone and vertebral column mineralization, while the *fbln1* mutant provides a hint that Fgf8 signaling controls the growth and morphogenesis of specific elements. Analyzing the detailed and probably various effects of the mutations on different regions of the zebrafish skeleton (head, skull, vertebrae, fins) will require more work in the future. 

## Figures and Tables

**Figure 1 biomolecules-14-00139-f001:**
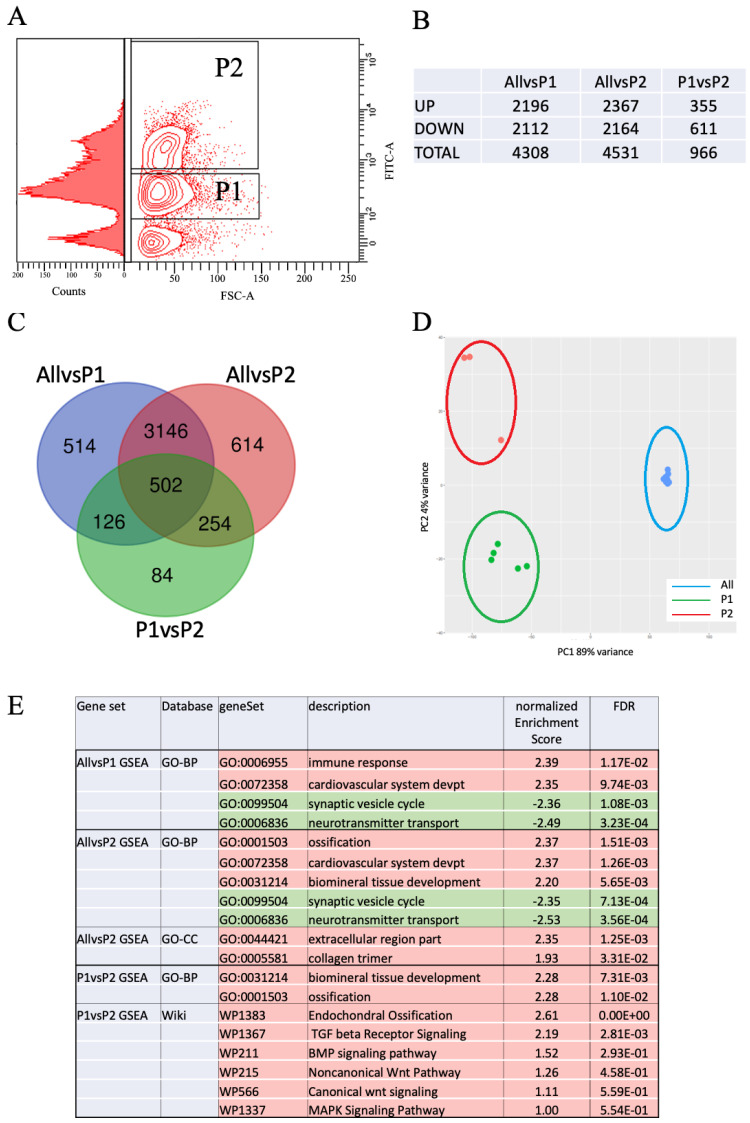
The osteoblasts at 4 dpf reveal two distinct populations based on the GFP fluorescence intensity and differential gene expression. (**A**) FACS plot showing forward scattering (FSC-A) and GFP fluorescence in singlet, living cells in gates P1 and P2. Cell distribution according to their GFP fluorescence is also shown to illustrate the two subpopulations P1 and P2. Gates were set to exclude the 100-fold larger population of non-fluorescent cells for illustration. (**B**) Number of DEGs that are up- or down-regulated in the different comparisons. (**C**) Venn diagram comparing the DEGs in “AllvsP1”, “AllvsP2”, and “P1vsP2”. (**D**) PCA plot, based on the 500 most variable genes in terms of normalized read counts in all individual samples, showing that the two cell subpopulations P1 and P2 are clearly different from each other, but very different from the whole larvae “All” population at 4 dpf. (**E**) Selected terms enriched in the DEG lists “AllvsP1”, “AllvsP2”, and “P1vsP2” as determined by GSEA analysis; columns represent the list concerned, the database used, the dataset concerned, its name, the normalized enrichment score, and the false discovery rate value (FDR). Positive enrichment scores indicate up-regulated terms (highlighted in red), negative ones refer to down-regulated terms (highlighted in green).

**Figure 2 biomolecules-14-00139-f002:**
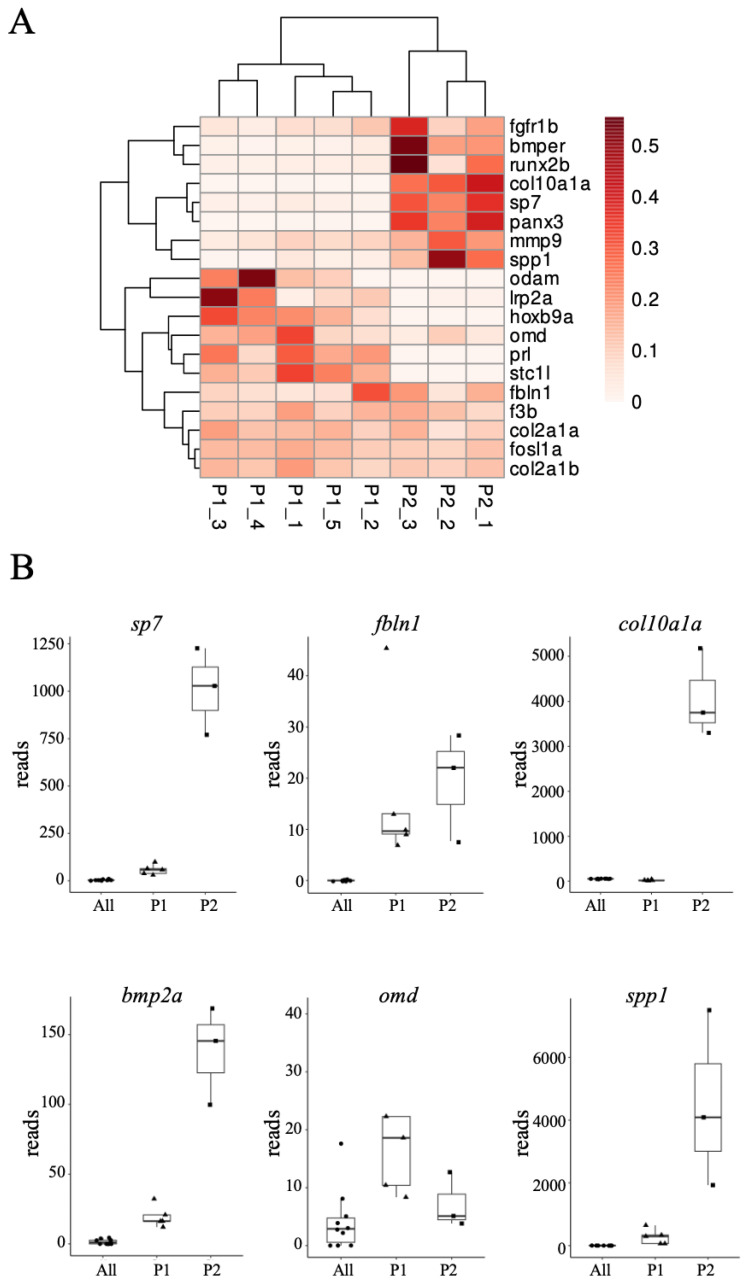
Specific genes present variable expression patterns when comparing the different cell populations. (**A**) change in the number of reads relative to the “All” population of selected genes; For each gene, the color code indicates the number of reads relative to the total number of reads for this gene in all the samples. (**B**) relative gene expression in the different cell populations. All samples are represented here: 10 for the “All” population, 5 for the P1, and 3 for the P2 subpopulations.

**Figure 3 biomolecules-14-00139-f003:**
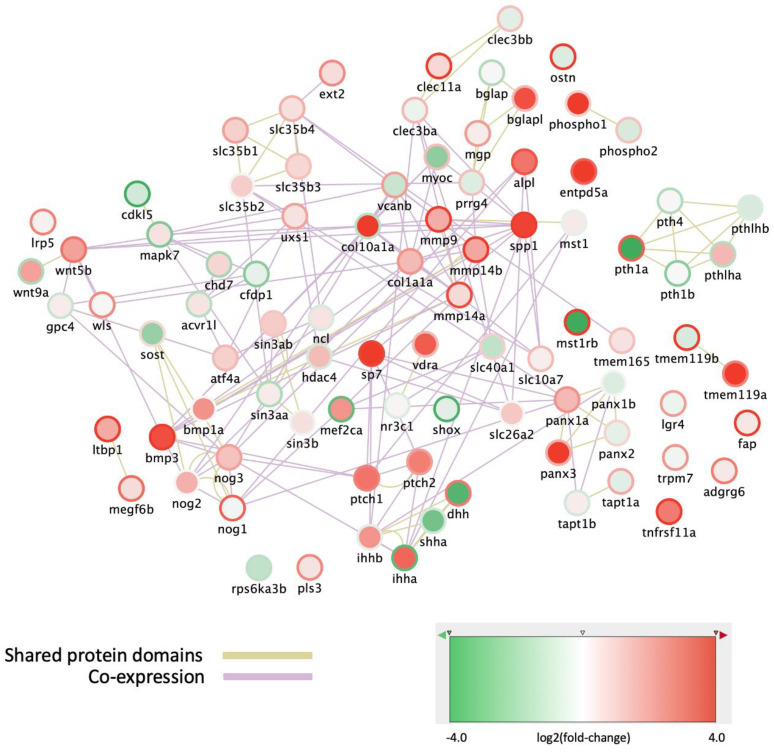
Expression changes in osteoblast subpopulations of genes involved in bone mineralization and ossification. The nodes represent genes, outer ring color represents the log(fold-change) between “All” to the P1 subpopulation, while the fill color represents the log(fold-change) between P1 and P2 subpopulations. The network was generated in Cytoscape, using the GeneMANIA databases for zebrafish Shared protein domains and Co-expression.

**Figure 4 biomolecules-14-00139-f004:**
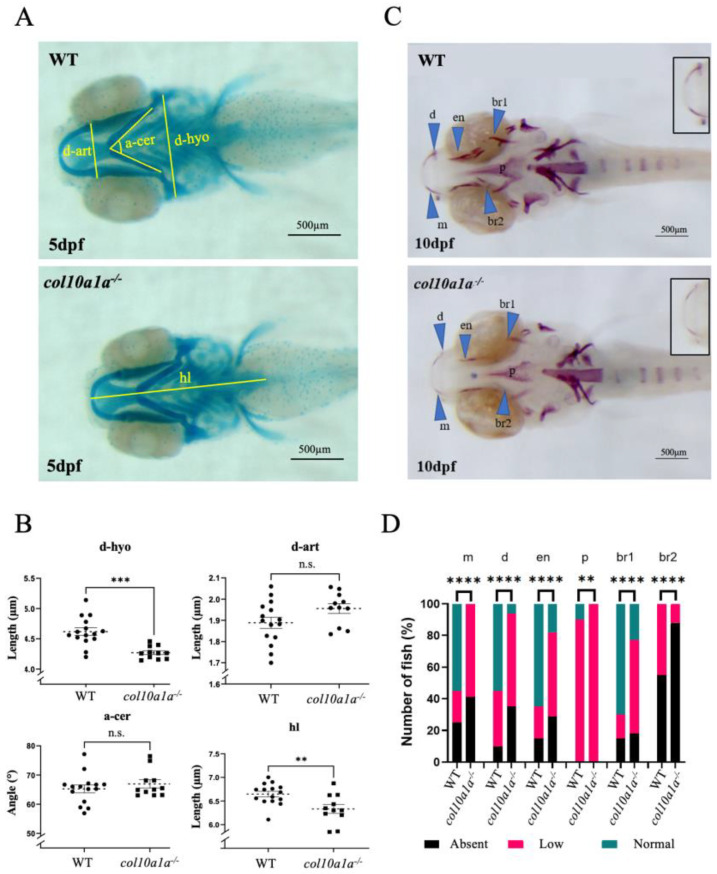
col10a1a^−/−^ mutants display a small chondrocranium at 5 dpf and decreased mineralization at 10 dpf compared to wt controls. (**A**) Ventral view of alcian blue stained WT and *col10a1a^−/−^* larvae at 5 dpf. The distances measured are indicated as described in Mat. and Meth. (**B**) *col10a1a^−/−^* reveal reduced head size at 5 dpf compared to WT (*p* < 0.01; WT n = 12, *col10a1a^−/−^* n = 15). Measures are distances from anterior to the posterior end of the ethmoid plate (head length-hl), between the two hyosymplectics (d-hyo), between the articulations joining the Meckel’s cartilage to the palatoquadrate (d-art), and the angle formed by the two ceratohyals (a-cer). (**C**) Ventral view of alizarin red stained WT and *col10a1a^−/−^* larvae at 10 dpf. The blue arrowheads point to the skeletal elements: maxillary (m), dentary (d), parasphenoid (p), entopterygoid (en), branchiostegal rays 1 and 2 (br1/br2). Inserts show the maxillary and dentary. (**D**) Fraction (%) of individuals presenting a high (green), reduced (red), or absent (black) level of bone mineralization in the different bone elements in WT and *col10a1a^−/−^* fish at 10 dpf. (WT n = 20, *col10a1a^−/−^* n = 17). (n.s. = non significant) significance: ** *p* < 0.01, *** *p* < 0.001, and **** *p* < 0.0001.

**Figure 5 biomolecules-14-00139-f005:**
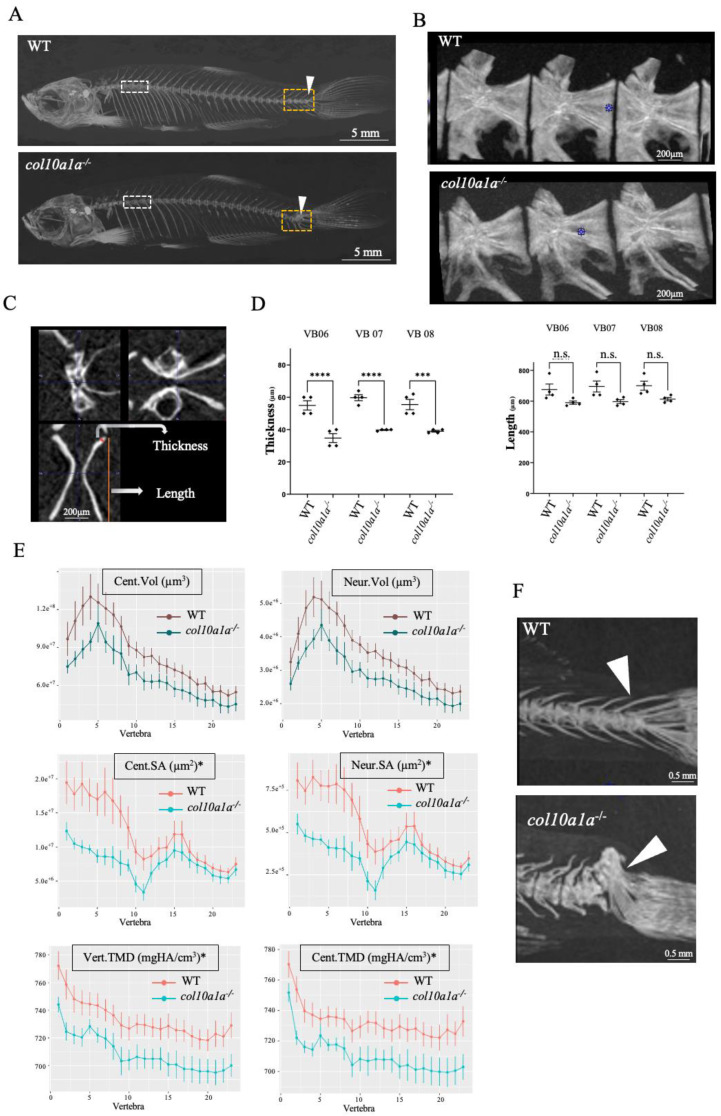
Adult *col10a1a^−/−^* zebrafish have decreased vertebral TMD and altered bone properties. (**A**) Representative µCT scans (MIPi = Maximum Intensity Projected image) of a year-old adult WT (top) and *col10a1a^−/−^* (bottom) reveal a decreased mineralization and fusion of the caudal fin vertebrae in the mutant. (**B**) Lateral view of pre-caudal vertebrae 6–8 (L to R) for WT and *col10a1a^−/−^* fish. (**C**) Representative µCT scan of a vertebra in 3 planar views, showing the two morphometric measurements: vertebral thickness (µm) and vertebral length (µm). (**D**) Comparison of morphometric measures on precaudal vertebrae 6–8 (n = 4 fish/group) in WT and *col10a1a^−/−^* fish. (**E**) Line plots generated using the FishCut software version 1.2 revealing significantly decreased TMDs in vertebra (Vert.TMD) and centra (Cent.TMD), as well as neural and centra surface areas (Neur.SA, Cent.SA) in *col10a1a^−/−^* relative to WT, while centra and neural arch volumes (Cent.Vol, Neur.Vol) were not significantly affected (* *p* < 0.05), (n = 6 fish/group). (**F**) *col10a1a^−/−^* mutants display fusion of the tail fin vertebra. (n.s. = non significant) significance: *** *p* < 0.001, and **** *p* < 0.0001.

**Figure 6 biomolecules-14-00139-f006:**
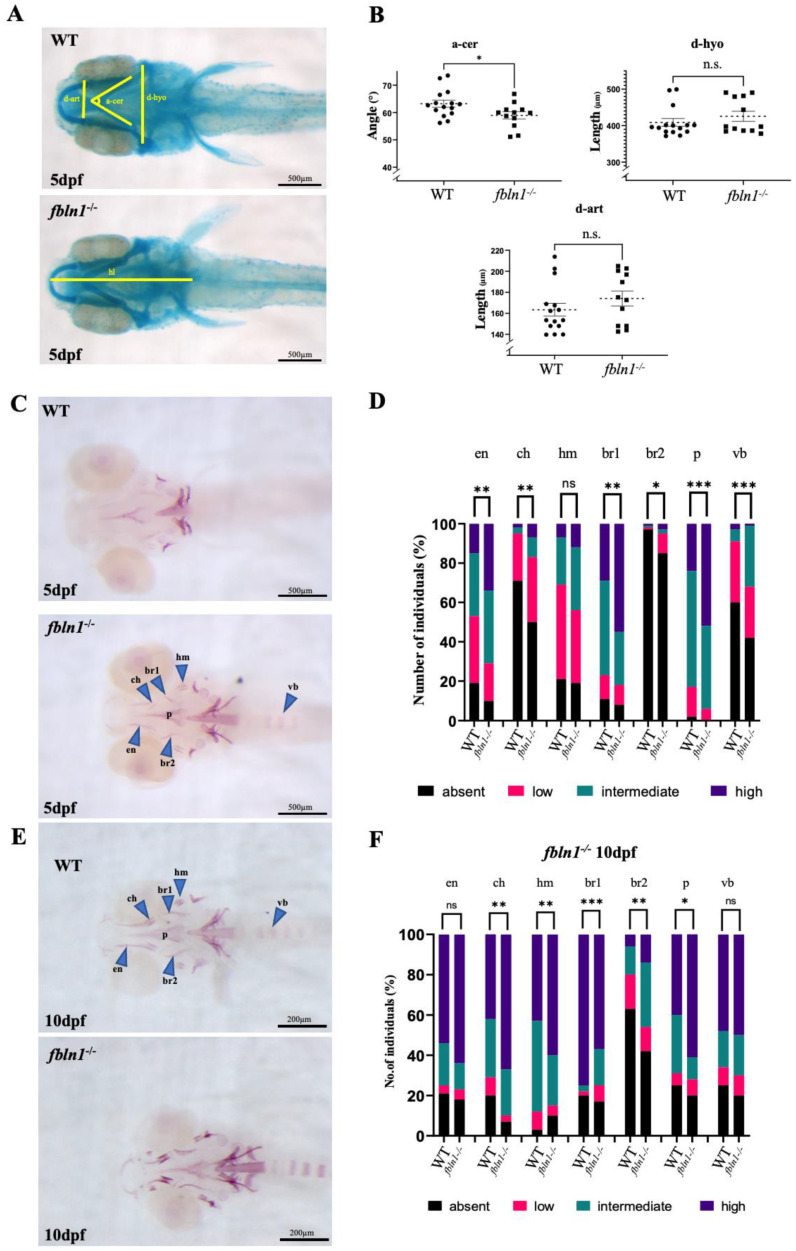
*fbln1^−/−^* mutants present increased mineralization at 5 dpf compared to WT. (**A**) Ventral view of alcian blue stained WT and *fbln1^−/−^* larvae at 5 dpf. The distances measured are indicated. (**B**) *fbln1^−/−^* reveal reduced angle between ceratohyals (a-cer) at 5 dpf compared to WT (WT n = 15, *fbln1*^−/−^ n = 12). (**C**) Ventral view of alizarin red stained WT and *fbln1^−/−^* larvae at 5 dpf. The blue arrowheads point to the skeletal elements: ceratohyal (ch), parasphenoid (p), entopterygoid (en), branchiostegal rays 1 and 2 (br1/br2), hyomandibular (hm), and vertebral body (vb). (**D**) Fraction (%) of individuals presenting a high (dark blue), intermediate (green), low (red), or absent (black) level of bone mineralization in the different bone elements in WT and *fbln1^−/−^* fish at 5 dpf. (WT n = 52, *fbln1^−/−^* n = 68). (**E**) Ventral view of alizarin red stained WT and *fbln1^−/−^* larvae at 10 dpf. (**F**) Fraction (%) of individuals presenting a high (dark blue), intermediate (green), low (red), or absent (black) level of bone mineralization in the different bone elements in WT and *fbln1^−/−^* fish at 10 dpf. (WT n = 35, *fbln1^−/−^* n = 40). (n.s. = non significant), significance: * *p* < 0.05, ** *p* < 0.01, and *** *p* < 0.001.

**Figure 7 biomolecules-14-00139-f007:**
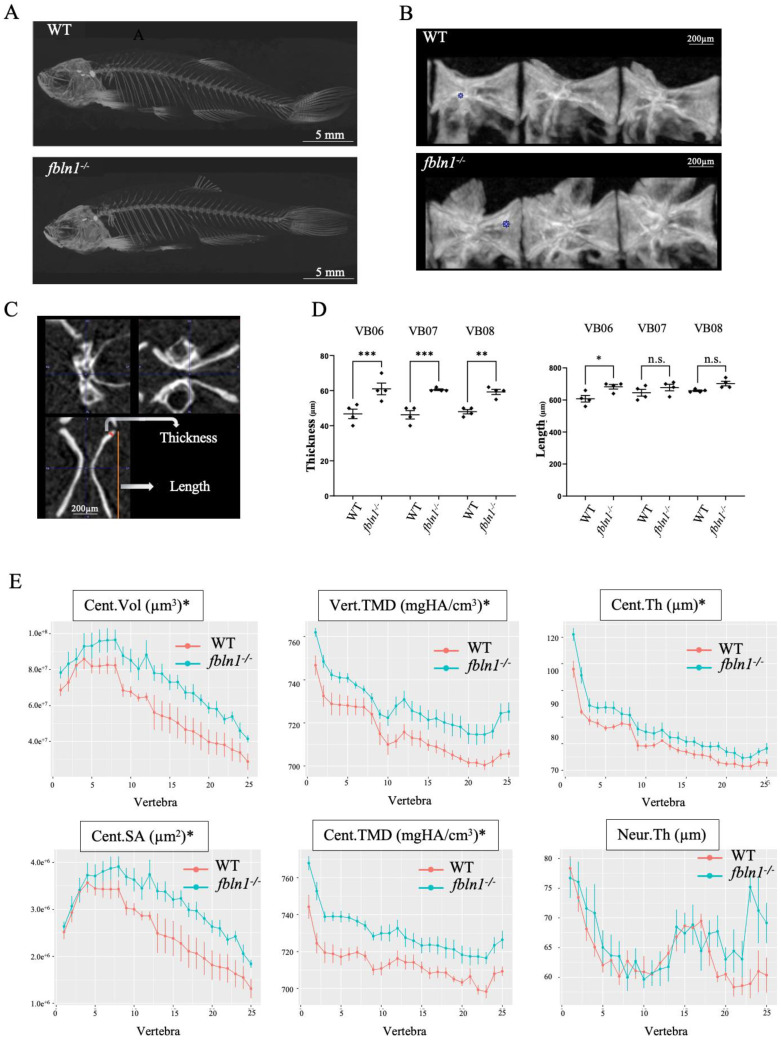
*fbln1^−/−^* adult zebrafish exhibit increased vertebral TMD and vertebral thickness. (**A**) µCT scans (MIPi = Maximum Intensity Projected image) of 1 year old adult WT and mutants. *fbln1^−/−^* larvae show an increased mineralization. (**B**) Lateral view of Vertebrae 6–8 (L to R) for WT and *fbln1^−/−^*, respectively. (**C**) Representative µCT scan of a vertebra in 3 planar views, showing two morphometric measurements: vertebral thickness (µm) and vertebral length (µm). (**D**) Morphometric analysis of individual precaudal vertebral body numbers 6–8 (n = 4 fish/group) revealed a significantly increased thickness and length of the vertebral body in *fbln1^−/−^* compared to WT controls. (**E**) Line plots generated using FishCuT software version 1.2 show significantly increased centra volume (Cent.Vol) and surface area (Cent.SA) in *fbln1^−/−^* compared to WT controls (n = 7 fish/group). Similarly, both vertebral (Vert.TMD) and centra TMD (Cent.TMD) are significantly increased in *fbln1^−/−^*. Significance: * *p* < 0.05, ** *p* < 0.01, and *** *p* < 0.001, n.s. = non-significant.

**Figure 8 biomolecules-14-00139-f008:**
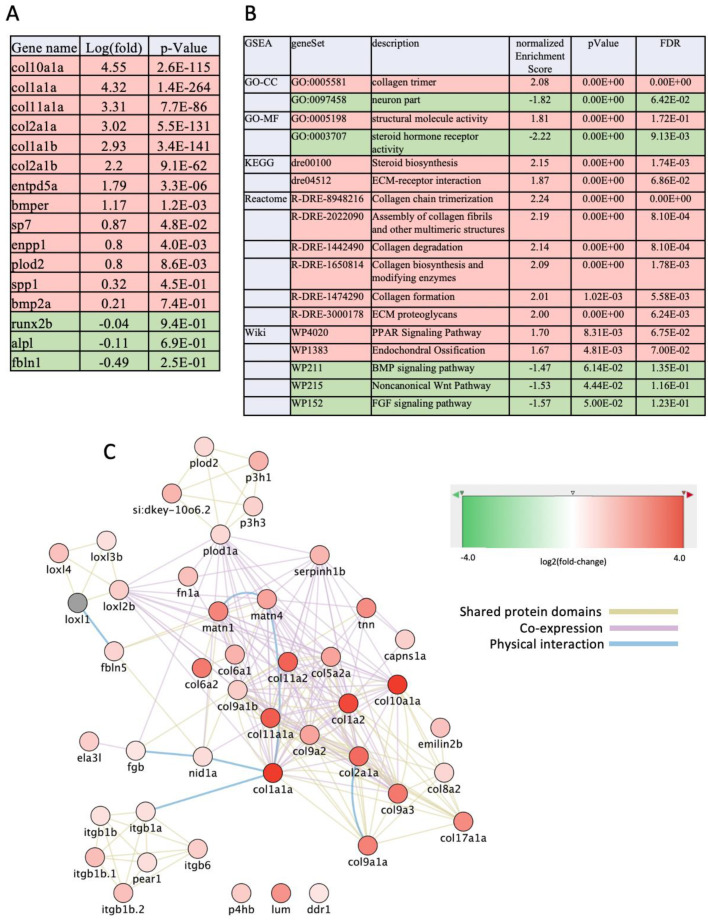
Changes of transcriptome in *fbln1^−/−^* mutants relative to WT at 10 dpf. (**A**) Log(fold-change) and *p*-values for some genes selected for their known role in osteogenesis. (**B**) Selected functional annotations of the list of DEGs in *fbln1^−/−^* mutants. (**C**) Network of DEGs centered around the genes involved in collagen biosynthesis. Red color refers to functions or genes that are upregulated, Green indicates genes or functions that are down-regulated in the mutants. In (**C**), the edges are color-coded according to the nature of interaction between nodes (genes) as indicated.

**Figure 9 biomolecules-14-00139-f009:**
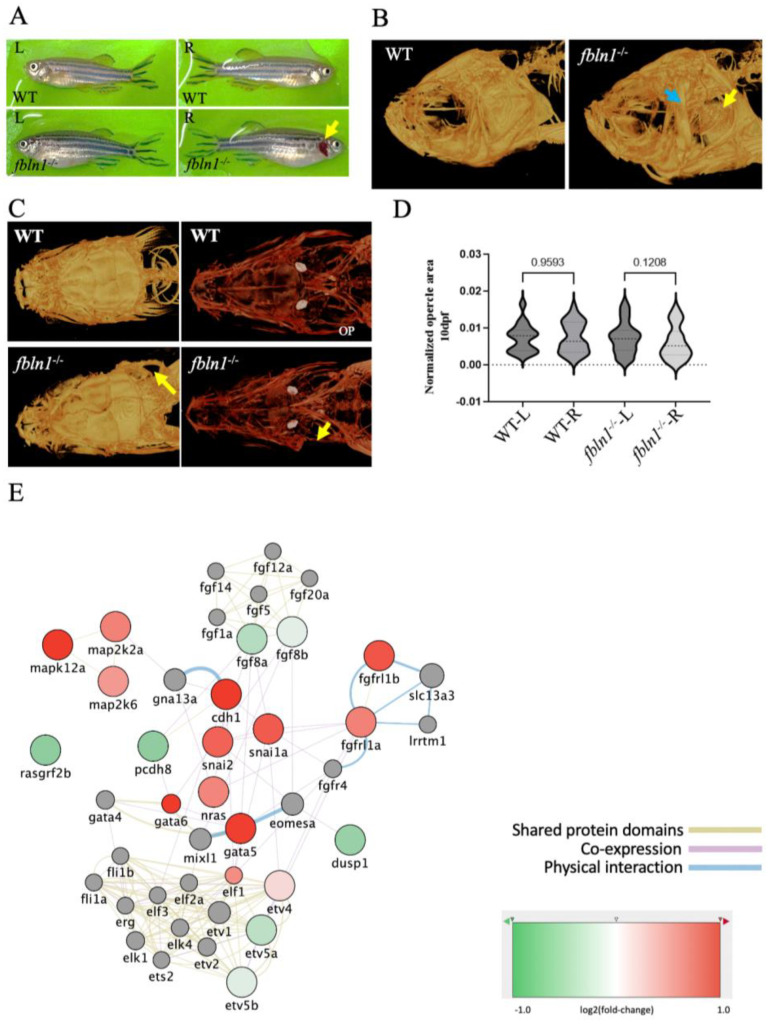
fbln1^−/−^ mutant zebrafish show missing opercle on the right side. (**A**) Adult *fbln1^−/−^* fish bright field image showing the missing opercle (yellow arrow). (**B**) 3D Reconstructed images from µCT scans of adult WT control and *fbln1^−/−^* fish head (yellow arrow: missing opercle, blue arrow: thickened subopercular bone). (**C**) 10 dpf Alizarin red stained zebrafish in ventral view (yellow arrow: missing opercle). (**D**) Left (L) and Right (R) opercle area measured in ventral view on 10 dpf alizarin red stained WT control and *fbln1^−/−^* zebrafish larvae show the trend of asymmetry between the L and R opercles in *fbln1^−/−^* mutant zebrafish (n = 34 WT, n = 37 *fbln1^−/−^* mutants). (**E**) Differentially expressed genes in *fbln1^−/−^* mutants that are involved in Fgf and Mapk signaling.

## Data Availability

The raw sequencing data are available at the Gene expression Omnibus (GEO) repository, respectively at https://www.ncbi.nlm.nih.gov/geo/query/acc.cgi?acc=GSE237934 (accessed on 5 January 2024) for the osteoblast sequencing and at https://www.ncbi.nlm.nih.gov/geo/query/acc.cgi?&acc=GSE238059 (accessed on 5 January 2024) for the *fbln1^−/−^* mutant.

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
