# Peer review of "The Osteoblast Transcriptome in Developing Zebrafish Reveals Key Roles for Extracellular Matrix Proteins Col10a1a and Fbln1 in Skeletal Development and Homeostasis"

_biomolecules, 2024, doi:10.3390/biom14020139_

Round 1
Reviewer 1 Report
Comments and Suggestions for Authors
The study is of interest, albeit solely descriptive. Proteomic analyses were not conducted on the proteins Col10a1a and Fbln1. It would be compelling to investigate the distribution of these two proteins in various tissues in both zebrafish larvae and adults. Are there other cell types expressing these markers?
Additionally, please specify the corresponding days post-fertilization (dpf) for the single-cell stage in which microinjection was performed (line 226).
Reviewer 2 Report
Comments and Suggestions for Authors
The manuscript by Raman et al. presents gene expression data related to two populations of osteoblasts identified using the zebrafish Sp7:GFP reporter line. It also presents the characterization of two zebrafish mutant lines for col10a1a and fbln1. While the new insights on the role of Col10a1a and Fbln1 in skeletal development and homeostasis provided by the manuscript are worth to be published, the manuscript failed to reach the quality necessary for publication. Many typos are present throughout the manuscript, non-SI units are used (e.g. hrs, uM, Mm), information is not homogenously presented and nomenclature for gene/protein names if not properly used, manuscript is too long (introduction should focus on osteoblasts only and should not contain results, methods should be more synthetic, results should present the essence of the expression data and discussion should not repeat that much the results), text in the figures are sometime hard to read and picture are too small, and English grammar can certainly be improved to enhance the readability. Conclusion: Data is interesting, but manuscript should be carefully revised to better present the results. My decision: major revisions.
More specific comments are included below:
Abstract
L25, what do you mean by the “status of signalling pathway is crucial”?
L26, simplify the sentence. You isolated Sp7+ osteoblasts using a fluorescent reporter.
L28-29, rephrase the first part of the sentence.
Introduction
L44, bone formation is the result of the deposition of bone material by osteoblasts and does not involve osteoclasts (bone resorbing cells). Do you mean bone remodelling, which involves both osteoblasts and osteoclasts? Since manuscript is about osteoblasts, I would remove text related to osteoclasts.
L61-62, vague please rephrase.
L68, what is under the tight control? Again, remove osteoclasts from your introduction.
L83, zebrafish operculum is also a skeletal structure commonly used to assess bone formation during early development.
L88, endomembranous??? It is intramembranous or endochondral ossification.
L101, osteogenesis imperfecta should not be italicized.
L106, gene acronyms are sometime completed with full name, other time they are not. Be homogenous.
L119-135, this summary of the results have nothing to do in the introduction.
Materials and Methods
L141, what do you mean by dry powder? Dry food, microdiet?
L147-151, be more synthetic on the description of the breeding.
L152 and throughout the text, add a space between unit and value.
L154, what do you mean by “the Tg positive larvae”?
L156, explain how fish were euthanized.
L162-163, remove redundant information already mentioned in previous sentence
L174, what is the meaning of 130-093-231?
L185, what is RNAsin? Define BSA.
L210, this paragraph is too long, remove non-essential information.
L211, what are the references ulg075 and ulg076?
L221, RT (I believe it is the acronym for room temperature) is not acceptable. Give a precise temperature.
L225, decimal separator in English is a period, not a comma.
L244, this paragraph is too long, remove non-essential information.
L246, indicate amount of MS-222. Define PFA
L269, remove redundant information regarding MS222. Define PBST
L274, AR or AR-S, homogenize throughout the manuscript.
L297, binocular?!?! Do you mean stereomicroscope?
L303, what do you mean by joing? What do you mean by Meckel’s?
L304, how did you estimate the degree of mineralization from images of AR-stained fish?
L311, overnight is not acceptable, give a precise duration. Define µCT.
L331, why this DOI?
Results
Expression data is presented in a very descriptive manner. Extract the essence and shorten this part.
L362, these populations of cells are certainly present in the siblings, they are just not labelled with GFP.
L363, genes and transcripts are different things!!! You did a whole transcriptome analysis not a whole genome analysis!
L380, they “may be” distinct osteoblasts.
L414, what do you mean by skeletal differentiation fate?
L434, Figure 2: why 5 replicates for P1 and only 3 for P2? Panel A, define units in the colour scale. Panel B, values in y axis are not readable. Title should be “Relative gene expression”
L442, what is a highly bone related gene?
L444, osteocalcin 1 not I
L445, what are relating enzymes?
L495, Figure 3: linking green and purple lines are barely visible in the graph. Rephrase first sentence of the caption. The presence of RANK (tnfrsf11a) in this network of gene should be further discussed.
L501-505, I am susrprised that the paper by Renn et al. published in 2013 in Developmental Biology (A col10a1:nlGFP transgenic line displays putative osteoblast precursors at the medaka notochordal sheath prior to mineralization) is totally overlooked here (and in the discussion).
L530, Figure 4: Panel B, enlarge letter size (use same font size throughout the figure). Homogenize information on p-value: give value or asterisks but do not mix. Panel C, define the inserts. Define more clearly in the figure caption the content of panel B and abbreviations.
L540, Figure 5: increase letter size and include scale bars! Panel E, increase graph size. Caption: -/- should be superscripted throughout the manuscript. Panel F and L570, I did not understand if the fusion of the tail fin vertebra is a single occurrence or if is a malformation commonly observed in the col1a1a mutants. If one occurrence, not sure it is worth to mention it. If common, then quantify the occurrence.
L553, into adulthood? Throughout will be better.
L573-576, rephrase.
L589, Figure 6. See previous comments on letter and image size, scale bars, etc.
L601, Figure 7. See previous comments on letter and image size, scale bars, etc.
L627, Figure 8. See previous comments on letter and image size, scale bars, etc. Panel B, define coloured arrows. Did you assess whether operculum defect could be gender related? Effect is not clear at 10 dpf, did you follow the effect over time?
L645, p=0.12 is definitely not a clear trend!!!!
L659-664, avoid the abundance of the adverb “clearly”
Discussion
L692-693, What do you mean by “very preferentially” and “very specifically”?
L698-703, avoid repetition of the results. Comment valid for the whole discussion.
L710, see previous comment on endomembranous?
L712, what is a perichondral osteoblast?
L722, Many?
L757, keeping?
L783-868, clearly identify the data used to draw these conclusions.
L891, the story around fgf8 is interesting but it is merely speculative and should not be over-analyzed as in the conclusion (L923). Are there other genes that could explain this observation?
Conclusion
L924-926: rephrase sentence.
L1083, authorship is the Gene Ontology Consortium
Comments on the Quality of English LanguageSee above
Reviewer 3 Report
Comments and Suggestions for Authors
In their publication “The osteoblast transcriptome in developing zebrafish reveals 2 key roles for extracellular matrix proteins Col10a1a and Fbln1 in skeletal development and homeostasis” identify Ratish Raman and coworkers, the expression pattern of two different zebrafish osteoblast populations and characterize two different knock-out models lacking the collagen x and fibulin protein. The authors have done quite a lot of experiments which show promising data. However I find it difficult to follow the overall aim of the publication. I think the experimental part is sufficient and clear but the structure of the paper needs careful reordering to make more sense.
I have marked minor problems.
Introduction
Something about fibulin 1 is missing
L190 30,000-50,000 cells
L298 maybe a crossreference or a separate image how you quantify would be nice to understand.
L353-499 This passage is very difficult to read and stay focused on. The nomenclature is very technical using P1 and P2 and in the further figures (Fig2 and 3) this is not specified. It would easier to call the P2 cells Osterix Positive or define as Osteoblasts something just not to search the whole paper to understand what P1 and P2 means.
L416 isn’t it obvious that SP7 is upregulated in cells FACS sortet for SP7? So doing interpretations or conclusions with this gene/protein are not really valid.
Fig4A/Fig6A either this is an overview how you have done the quantification or you are comparing the wild type with the collagen ko. In the first case, one image of unspecified genotype is sufficient in the second case you should mark the same structures so that the reader can be convinced that there are differences.
Fig4D which statistical test has been used, since these is a scoring a t-test wouldn’t be right?
L573ff I would put this information into the introduction, as well
The discussion is confusing, much is repetition and clarification of the results and not interpretation or integration of the results in the existing literature. Also it is not clear if there is a connection between collagen 10 and fibulin and why genes of these proteins are taken and put together as a knock out strategy.
Also the finding of the opercle lacking in the fibulin deficient animal is a nice finding and somehow not integrated into the overall flow of the paper.
As stated above I think the experimental data is sufficient, but the authors should take some work to reorganize the paper to put them into a fitting train of thought. Especially the discussion and the introduction need work.
Round 2
Reviewer 1 Report
Comments and Suggestions for Authors
The authors have adressed my questions and the paper has been improved.
Reviewer 2 Report
Comments and Suggestions for Authors
The revised manuscript reads better than the original manuscript but there are still many typos that must be corrected. I urge all the 18 co-authors to carefully proof-read the manuscript. Some comments were also not properly addressed (see below).
L115-124: Again, the last part of the introduction is not the place to summarize the findings of the manuscript nor to conclude on them. Identify the scientific question(s) to be answered, then present your objectives/approach to answer the question(s).
L144: by adding 0.048% (w/v) of MS-222
L149: define HBSS
L160: authors identified 130-093-231 as an order number and said it was deleted. It is not.
L177: authors said RNAsin was defined. It is not.
L282-283: Authors did not answer my previous comment: how the degree of mineralization was estimated from images of AR-stained fish? Did you infer from pixel intensity? Did you use an ImageJ macro? The structure of the sentence is also weird: “bone (alizarin red) images were evaluated by estimating the degree of mineralization” would read better as “degree of mineralization was estimated from images of alizarin red stained bone structures”.
L306-307: gene names should be italicized. Check the rest of the manuscript for gene/protein name nomenclature (e.g. osterix at L336 and bmp3, bmp1a and ltbp1 at L421, etc.)
L341-341: this sentence is still weird. The statement that the two osteoblast populations – identified in the reporter line based on GFP fluorescence – are “not observed” in the wild-type line is still misleading. These sp7-positive cells are most probably present in wild-type fish, but they are not labelled with GFP, thus they cannot be sorted by FACS. On the contrary, the absence of fluorescence signal in wild-type fish indicates that cells sorted by FACS in fish of the reporter line are truly sp7-positive cells and not cells with autofluorescence that would also be present in wild-type fish.
L353: not clear from which subpopulation of genes (i.e. ALLvsP1, ALLvsP2, ALLvsP1&P2 or V1vsP2) those 500 are related to. Please add this information.
L384: what do you mean by “near perfect enrichment”?
L389: what do you mean by “skeletal differentiation”? Cells such as osteoblasts can differentiate, but skeleton?!?!
L404: Authors answered to my comment on the numbers of replicates saying “Obviously, the lower number of cells in the P2 subpopulation led to low quality RNA in some of the samples”. I disagree, RNA quality is not related to the number of cells.
L410-452: I could not see the “reorganization, simplification, and shortening” of the expression data claimed by the authors. Text is still very descriptive and not fluid…
L495: Are the inserts in panel C really needed? There are not much bigger that in the original images and do not bring additional information.
L565: Homogenize p-value information in panel D; use asterisks or n.s. Homogenize superscripted -/- throughout the figure.
L615 and L636: I still do not agree with panel D and the conclusions taken from it. Statistical significance was set at p<0.05 (L329-330), thus p-values depicted in panel D indicate that differences between left and right opercular bones are not significant in both WT and mutant lines. A similar comment applies to p-values depicted in L649-650; p=0.6 and p=0.44 indicate no significant differences.
L698: what type of cells are you referring to?
L699: to be seen? Please rephrase.
L716-718: indicate in which species these observations were made. Ref 24 is in medaka. Ref 77 is in zebrafish, correct?
L724-730: this paragraph is clearly (re)presenting results.
L754: “the most striking such defect”? Please rephrase.
L755-756: not supported by statistical analysis. Please rephrase or provide data that demonstrates a significant reduction in operculum area.
L765: “may lead to a downregulation”.
L850: reference list has many inconstancies (journal names are abbreviated or not, capitalized or not, some doi numbers are duplicated, page/article numbers are missing, etc.). It should be corrected.
Comments on the Quality of English Languagesee above
Reviewer 3 Report
Comments and Suggestions for Authors
The revised study is fine with me, every issue from my side has been addressed.
